# Learning to Compose Representations of Different Encoder Layers towards Improving Compositional Generalization

**Lei Lin[1,2]\*** **Shuangtao Li[1,2]\*** **Yafang Zheng[1,2]\*** **Biao Fu[1,2]** **Shan Liu[1,2]**
**Yidong Chen[1,2]** **Xiaodong Shi[1,2]†**

[1]Department of Artificial Intelligence, School of Informatics, Xiamen University
[2]Key Laboratory of Digital Protection and Intelligent Processing of Intangible Cultural Heritage
of Fujian and Taiwan (Xiamen University), Ministry of Culture and Tourism, China
{linlei}@stu.xmu.edu.cn,{ydchen,mandel}@xmu.edu.cn

## Abstract

Recent studies have shown that sequence-to-sequence (seq2seq) models struggle with compositional generalization (CG), i.e., the ability to systematically generalize to unseen compositions of seen components. There is mounting evidence that one of the reasons hindering CG is the representation of the encoder uppermost layer is entangled, i.e., the syntactic and semantic representations of sequences are entangled. However, we consider that the previously identified representation entanglement problem is not comprehensive enough. Additionally, we hypothesize that the source keys and values representations passing into different decoder layers are also entangled. Starting from this intuition, we propose COMPOSITION (**Compo**se **S**yntactic and Seman**t**ic Representa**tion**s), an extension to seq2seq models which learns to compose representations of different encoder layers dynamically for different tasks, since recent studies reveal that the bottom layers of the Transformer encoder contain more syntactic information and the top ones contain more semantic information. Specifically, we introduce a *composed layer* between the encoder and decoder to compose different encoder layers' representations to generate specific keys and values passing into different decoder layers. COMPOSITION achieves competitive results on two comprehensive and realistic benchmarks, which empirically demonstrates the effectiveness of our proposal. Codes are available at https://github.com/thinkaboutzero/COMPOSITION.

## 1 Introduction

A crucial property of human language learning is its *compositional generalization* (CG) — the algebraic ability to understand and produce a potentially infinite number of novel combinations from known components (Fodor and Pylyshyn, 1988; Lake et al.,

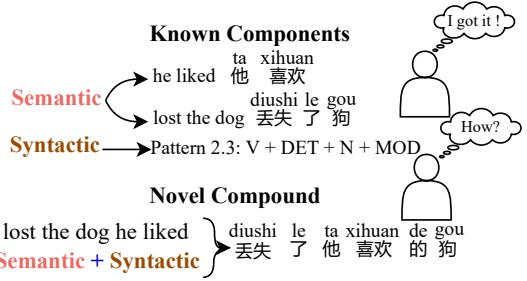

Figure 1: Examples from CoGnition (Li et al., 2021) show the workflow of how humans exhibit CG. Suppose interpreters know the translation: [丢失了狗] for "lost the dog" and [他喜欢] for "he liked" (semantic information). When they first encounter "lost the dog he liked", they can correctly translate [丢失了他喜欢的狗] instead of [丢失了狗他喜欢] depending on Pattern 2.3 (syntactic information).

2017). For example, if a person knows "the doctor watches a movie" [医生看电影][1] and "the lawyer" [律师], then it is natural for the person to know the translation of "the lawyer watches a movie" is [律师看电影] even though they have never seen it before. Such nature is beneficial for generalizing to new compositions of previously observed elements, which is often required in real-world scenarios.

Despite astonishing successes across a broad range of natural language understanding and generation tasks (Sutskever et al., 2014; Dong and Lapata, 2016; Vaswani et al., 2017), neural network models, in particular the very popular sequence-to-sequence (seq2seq) architecture, are argued difficult to capture the compositional structure of human language (Lake and Baroni, 2018; Keysers et al., 2020; Li et al., 2021). A key reason for failure on CG is different semantic factors (e.g., lexical meaning and syntactic patterns) required by CG are entangled, which was proved explicitly or implicitly to exist in the representation of the encoder uppermost layer (encoder entanglement

---

\*Equal Contribution.
†Corresponding Author.

[1]The sentence in "[]" denotes the Chinese translation.

problem) by previous studies (Li et al., 2019; Raunak et al., 2019; Russin et al., 2019; Liu et al., 2020b, 2021; Jiang and Bansal, 2021; Zheng and Lapata, 2022a; Yin et al., 2022; Ruis and Lake, 2022; Li et al., 2022; Cazzaro et al., 2023). In other words, the syntactic and semantic representations of sequences are entangled.

In order to alleviate the encoder entanglement problem, one line of research on CG mainly concentrate on improving the encoder representation or separating the learning of syntax and semantics which adopt similar approaches to humans' strategies for CG (see Figure 1). Specifically, several works either produce two separate syntactic and semantic representations, and then compose them (Li et al., 2019; Russin et al., 2019; Jiang and Bansal, 2021) or design external modules, and then employ a multi-stage generation process (Liu et al., 2020b, 2021; Ruis and Lake, 2022; Li et al., 2022; Cazzaro et al., 2023). Moreover, some studies explore bag-of-words pre-training (Raunak et al., 2019), newly decoded target context (Zheng and Lapata, 2022a,b) or prototypes of token representations over the training set (Yin et al., 2022) to improve the encoder representation. Furthermore, we hypothesize that the source keys and values representations passing into different decoder layers are also entangled (keys, values entanglement problem), not just the representation of the encoder uppermost layer. We will further illustrate it in Section 5.1.

Therefore, one natural question can be raised: how to alleviate keys, values entanglement problem. As a remedy, we examine CG from a new perspective to solve it, i.e., utilizing different encoder layers' information. We conduct preliminary analysis provided in Appendix A, and conclude that the bottom layers of the Transformer encoder contain more syntactic information and the top ones contain more semantic information. Inspired by this, we collect representations outputted by each encoder layer instead of separating the learning of syntax and semantics. So one intuitive solution to solve keys, values entanglement problem is to learn different and specific combinations of syntactic and semantic information (i.e., representations outputted by each encoder layer) for keys and values of different decoder layers. We argue that *an effective composition is to provide different combinations for different tasks and a specific combination for a particular task*. For example, the model can learn preference of layers in different levels

of the encoder for different tasks (i.e., For A task, the information at encoder layer 0 may be more important, however, for B task, the information at encoder layer 5 may be more important). Additionally, the model can select which encoder layer of information is most suitable for itself (that is, which encoder layer of information is the most important) for a particular task. Inspired by that, we propose *the composed layer* (learnable scalars or vectors) to generate different specific source keys and values passing into different decoder layers for different particular tasks, since we argue that the learned scalars or vectors (i.e., different dynamic composition modes) by the model itself during training process can be dynamically adjusted for different particular tasks, and provide a way to learn preference of layers in different levels of the encoder for a particular task. Putting everything together, we propose COMPOSITION (**Compo**se **S**yntactic and **S**emantic Representa**tion**s), an extension to seq2seq models that learns to compose the syntactic and semantic representations of sequences dynamically for different tasks. COMPOSITION is simple yet effective, and mostly applicable to any seq2seq models without any dataset or task-specific modification.

Experimental results on CFQ (Keysers et al., 2020) (semantic parsing) and CoGnition (Li et al., 2021) (machine translation, MT) empirically show that our method can improve generalization performance, outperforming competitive baselines and other techniques. Notably, COMPOSITION achieves **19.2%** and **50.2%** (about **32%**, **20%** relative improvements) for instance-level and aggregate-level error reduction rates on CoGnition. Extensive analyses demonstrate that composing the syntactic and semantic representations of sequences dynamically for different tasks leads to better generalization results.

## 2 Related Work

**Compositional Generalization.** After realizing existing neural models still struggle in scenarios requiring CG (Lake and Baroni, 2018; Keysers et al., 2020; Li et al., 2021), there have been various studies attempt to improve the model's ability of CG, including data augmentation (Andreas, 2020; Akyürek et al., 2021; Yang et al., 2022; Li et al., 2023), modifications on model architecture (Li et al., 2019; Russin et al., 2019; Nye et al., 2020; Liu et al., 2020c, 2021; Zheng and Lapata, 2021;

Herzig and Berant, 2021; Chaabouni et al., 2021; Mittal et al., 2022; Zheng et al., 2023), intermediate representations (Furrer et al., 2020; Herzig et al., 2021), meta-learning (Lake, 2019; Conklin et al., 2021), explorations on pre-trained language models (Furrer et al., 2020; Zhou et al., 2023), auxiliary objectives (Jiang and Bansal, 2021; Yin et al., 2023), two representations (Li et al., 2019; Russin et al., 2019; Jiang and Bansal, 2021) and enriching the encoder representation (Raunak et al., 2019; Zheng and Lapata, 2022a,b; Yin et al., 2022; Yao and Koller, 2022). One line of research exploring how to alleviate the encoder entanglement problem has attracted much attention. Our work is in line with it, but we examine CG from a new perspective, i.e., utilizing different encoder layers' information.

**Neural Machine Translation.** Recently, CG and robustness of Neural Machine Translation (NMT) have gained much attention from the research community (Cheng et al., 2020; Xu et al., 2021; Lake and Baroni, 2018; Li et al., 2021), including pre-training (Raunak et al., 2019), data augmentation (Guo et al., 2020a), datasets (Li et al., 2021), and enriching semantic information at token-level (Thrush, 2020; Akyurek and Andreas, 2021; Zheng and Lapata, 2022a,b; Yin et al., 2022). Noteworthily, Dankers et al. (2022) argue that MT is a suitable and relevant testing ground to test CG in natural language. Different from them, we introduce a composed layer to compose different encoder layers' information dynamically, which is inspired by previous studies about analyzing Transformer (Raganato et al., 2018; Voita et al., 2019).

**Encoder Layer Fusion.** Encoder layer fusion (EncoderFusion) is a technique to fuse all the encoder layers (instead of the uppermost layer) for seq2seq models, which has been proven beneficial, such as layer attention (Bapna et al., 2018; Shen et al., 2018; Wang et al., 2019), layer aggregation (Dou et al., 2018; Wang et al., 2018; Dou et al., 2019), and layer-wise coordination (He et al., 2018; Liu et al., 2020a). However, other studies show that exploiting low-layer encoder representations fails to improve model performance (Domhan, 2018). The essence of different EncoderFusion works is to explore different ways to combine information from different encoder layers. Our approach is essentially the same as EncoderFusion work, which explores different ways to combine information from different encoder layers, however, *we propose a new way to combine them*. Meanwhile, we

consider that there are also three distinct differences. **Firstly**, our method exploits information from all encoder sub-layers and generates specific keys, values passing into different decoder layers while they do not. **Secondly**, our method shows the effectiveness of utilizing low-layer encoder representations while they have the opposite view (see Appendix D). **Thirdly**, we do not share the same motivation or task. Their work focuses on how to transform information across layers in deep neural network scenarios for seq2seq tasks. Our motivation is to compose the syntactic and semantic representations of sequences dynamically for CG.

## 3 Methodology

We adopt the Transformer architecture (Vaswani et al., 2017) to clarify our method, however, **our proposed method is mostly applicable to any seq2seq models**. In the following, we first introduce the Transformer baseline (Section 3.1), and then our proposed COMPOSITION (Section 3.2).

### 3.1 Transformer

The Transformer (Vaswani et al., 2017) is designed for sequence to sequence tasks which adopts an encoder-decoder architecture. The multi-layer encoder summarizes a source sequence into a contextualized representation and another multi-layer decoder produces the target sequence conditioned on the encoded representation.

Formally, given a sequence of source sentence $X = \{x_1, \ldots, x_S\}$ and a sequence of target sentence $Y = \{y_1, \ldots, y_T\}$, where $S, T$ denote the number of source and target tokens, respectively. $\mathcal{D} = \{(X, Y), \ldots\}$ denotes a training corpus, $\mathcal{V}$ denotes the vocabulary of $\mathcal{D}$, and $\theta$ denotes parameters of the Transformer model. The model aims to estimate the conditional probability $p(y_1, \ldots, y_T | x_1, \ldots, x_S)$:

$$p(Y|X; \theta) = \prod_{t=1}^{T+1} p(y_t | y_{<t}, X; \theta), \quad (1)$$

where $t$ is the index of each time step, $y_{<t}$ denotes a prefix of $Y$ and each factor $p(y_t | X, y_1, \ldots, y_{t-1}; \theta)$ is defined as a $softmax$ distribution of $\mathcal{V}$.

During training, the model are generally optimized with the cross-entropy (CE) loss, which is calculated as follows:

$$L_{CE}(\theta) = -\sum_{t=1}^{T+1} \log p(y_t | y_{<t}, X; \theta). \quad (2)$$

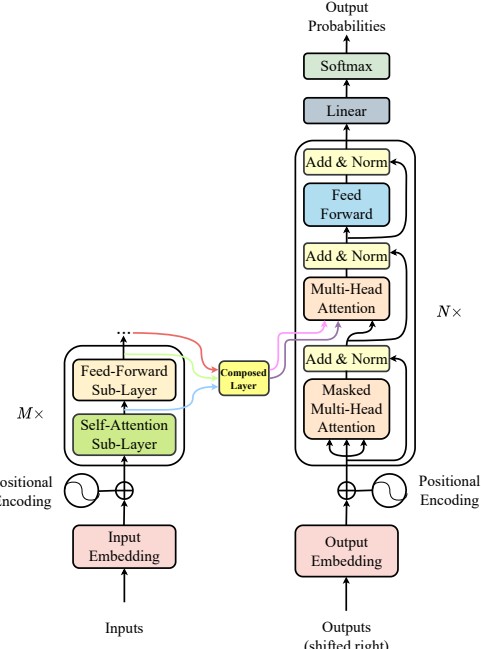

the Transformer encoder first maps $X$ to embeddings $H_0$, and then $H_0$ are fed into a Transformer self-attention sub-layer and feed-forward sub-layer to generate $H_1^{SA} \in \mathbb{R}^{d \times S}, H_1^{FF} \in \mathbb{R}^{d \times S}$ respectively, where $d$ denotes the hidden size. Next, each subsequent encoder layer takes the previous layer's output as input. The overall process is as follows:

$$H_1^{SA} = f_{Self-Attention}(H_0), \qquad (3)$$

$$H_1^{FF} = f_{Feed-Forward}(H_1^{SA}), \qquad (4)$$

$$H_i^{SA} = f_{Self-Attention}(H_{i-1}^{FF}), \qquad (5)$$

$$H_i^{FF} = f_{Feed-Forward}(H_{i-1}^{SA}), \qquad (6)$$

where $2 \leq i \leq M$ denote $i$-th encoder layer. Therefore, we can collect representations outputted by each encoder sub-layer $H_{collect} = \{H_1^{SA}, H_1^{FF}, \ldots, H_M^{SA}, H_M^{FF}\}$. The keys and values of multi-head attention module of decoder layer $l$ are defined to be:

$$H_{key}^l = \sum_{i=1}^{2M} w_k^i H_{collect}^i, \qquad (7)$$

$$H_{value}^l = \sum_{i=1}^{2M} w_v^i H_{collect}^i, \qquad (8)$$

where $w_k^i \in \mathbb{R}, w_v^i \in \mathbb{R}$ are learnable scalars or vectors and mutually different (e.g. $w_k^i \neq w_v^i$, $w_k^i \neq w_k^j$ and $w_v^i \neq w_v^j$), which weight each collected source representation in a dynamic linear manner. Eq. 7 and 8 provide a way to learn preference of sub-layers in different levels of the encoder.

## 4 Experiments

We mainly evaluate COMPOSITION on two comprehensive and realistic benchmarks for measuring CG, including *CFQ* (Keysers et al., 2020) and *CoGnition* (Li et al., 2021).

### 4.1 Experimental Settings

**Datasets.** *CoGnition* is a recently released realistic English → Chinese (En→Zh) translation dataset, which is used to systematically evaluate CG in MT scenarios. It consists of a training set of 196,246 sentence pairs, a validation set and a test set of 10,000 samples. In particular, it also has a dedicated synthetic test set (i.e., CG-test set) consisting of 10,800 sentences containing novel compounds, so that the ratio of compounds that are correctly translated can be computed to evaluate the model's

Figure 2: Architecture of COMPOSITION based on the Transformer. The bright yellow block in the middle denotes the composed layer introduced in Section 3.2. The red line denotes that we collect representations of the same positions for the rest encoder layers.

During inference, the model predicts the probabilities of target tokens in an auto-regressive mode and generates target sentences using a heuristic search algorithm, such as beam search (Freitag and Al-Onaizan, 2017).

## 3.2 COMPOSITION

Our proposed COMPOSITION extends the Transformer by introducing a composed layer between the encoder and decoder. Figure 2 shows the overall architecture of our approach.

### 3.2.1 Composed Layer

The composed layer is a list consisting of $2N$ learnable vectors due to $2N$ source keys, values passing into $N$ decoder layers, where each vector involves $2M$ learnable scalars or vectors. $M, N$ denote the number of encoder and decoder layers respectively.

### 3.2.2 Dynamic Combination

Here, we describe how to use the composed layer to compose collected representations dynamically for generating specific keys and values representations passing into different decoder layers. Let $f_{Self-Attention}$ and $f_{Feed-Forward}$ denote a Transformer self-attention sub-layer and feed-forward sub-layer respectively. The embedding layer of

| Model | #Params | Compound Translation Error Rate (CTER) ↓ | | | | | BLEU ↑ |
|---|---|---|---|---|---|---|---|
| | | NP | VP | PP | Total | Δ | |
| Transformer | 35M | 24.7%/55.2% | 24.8%/59.5% | 35.7%/73.9% | 28.4%/62.9% | -/- | 59.5 |
| Transformer-Rela | 35M | 30.1%/58.1% | 27.6%/61.2% | 38.5%/74.1% | 32.1%/64.5% | +3.7%/+1.6% | 59.1 |
| Transformer-Small | 25M | 25.1%/56.9% | 25.6%/60.3% | 39.1%/75.0% | 29.9%/64.5% | +1.5%/+1.6% | 59.0 |
| Transformer-Deep | 40M | 23.3%/51.6% | 24.1%/58.0% | 33.8%/72.6% | 27.0%/60.7% | -1.4%/-2.0% | 60.1 |
| Bow | 35M | 22.2%/47.9% | 24.8%/55.6% | 35.0%/73.2% | 27.3%/58.9% | -1.1%/-3.0% | - |
| SeqMix | 35M | 24.5%/49.7% | 26.9%/58.9% | 34.4%/73.1% | 28.6%/60.6% | +0.2%/-2.3% | - |
| Dangle | 35M | -/- | -/- | -/- | 24.4%/55.5% | -5.0%/-7.4% | 59.7 |
| Proto-Transformer | 42M | 14.1%/36.5% | 22.1%/50.9% | 28.9%/68.2% | 21.7%/51.8% | -6.7%/-11.1% | 60.1 |
| Transformer+CReg | 25M | -/- | -/- | -/- | 20.2%/48.3% | -8.2%/-14.6% | 61.3 |
| R-Dangle$_{sep}$ | 70M | -/- | -/- | -/- | **16.0%/42.1%** | **-12.4%/-20.8%** | **63.4** |
| DLCL | 35M | -/- | -/- | -/- | 28.4%/67.9% | +0.0%/+5.0% | 59.2 |
| COMPOSITION | 35M | **10.0%/32.6%** | 22.1%/54.8% | 29.2%/68.5% | 20.4%/52.0% | -8.0%/-10.9% | 61.5 |
| COMPOSITION-Rela | 35M | 15.5%/39.2% | 22.4%/54.0% | 29.1%/67.3% | 22.3%/53.5% | -6.1%/-9.4% | 61.6 |
| COMPOSITION-Small | 25M | 14.3%/40.3% | 24.4%/58.1% | 34.5%/73.4% | 24.4%/57.3% | -4.0%/-5.6% | 60.1 |
| COMPOSITION-Deep | 40M | 11.4%/34.7% | **19.5%/50.4%** | **26.7%/65.6%** | 19.2%/50.2% | -9.2%/-12.7% | 62.0 |

Table 1: CTERs (%) on CoGnition. We report instance-level and aggregate-level CTERs in the CG-test set, separated by "/". In addition, we also report the commonly used metric BLEU score in MT tasks. "-" denotes that the results are not provided in the original paper. Results are averaged over 6 random runs.

| CFQ | |
|---|---|
| **x** | "Did M0 direct M1" |
| **y** | SELECT count ( * ) WHERE { M0 film.director.film M1 } |

| CoGnition | |
|---|---|
| **x** | "the dog he liked practiced all weekend long ." |
| **y** | 他 喜欢 的 狗 整个 周末 都 在 练习 。 |

Figure 3: Examples of CFQ and CoGnition.

ability of CG directly. *CFQ* is automatically generated from a set of rules in a way that precisely tracks which rules (atoms) and rule combinations (compounds) of each example. In this way, we can generate three splits with *maximum compound divergence* (MCD) while guaranteeing a small atom divergence between train and test sets, where large compound divergence denotes the test set involves more examples with unseen syntactic structures. We evaluate our method on all three splits. Each split dataset consists of a training set of 95,743, a validation set and a test set of 11,968 examples. Figure 3 shows examples of them.

**Data Preprocess.** We follow the same settings of Li et al. (2021) and Keysers et al. (2020) to pre-process CoGnition and CFQ datasets separately. For CoGnition, we use an open-source Chinese to-kenizer[2] to preprocess Chinese and apply Moses tokenizer[3] to preprocess English, which is the same in Lin et al. (2023) and Liu et al. (2023). We employ byte-pair encoding (BPE) (Sennrich et al., 2016) for Chinese with 3,000 merge operations, generating a vocabulary of 5,500 subwords. We do not apply BPE for English due to the small vocabulary (i.e., 2000). For CFQ, we use the GPT2-BPE

tokenizer[4] to preprocess source and target English text.

**Setup.** For CoGnition and CFQ, we follow the same experimental settings and configurations of Li et al. (2021) and Zheng and Lapata (2022a) respectively. We implement all comparison models and COMPOSITION with an open source Fairseq toolkit (Ott et al., 2019). More details are provided in Appendix B.

**Evaluation Metrics.** For CoGnition, we use compound translation error rate (CTER (Li et al., 2021)) to measure the model's ability of CG. Specifically, *instance-level* CTER denotes the ratio of samples where the novel compounds are translated incorrectly, and *aggregate-level* CTER denotes the ratio of samples where the novel compounds suffer at least one incorrect translation when aggregating all 5 contexts. To calculate CTER, Li et al. (2021) manually construct a dictionary for all the atoms based on the training set, since each atom contains different translations. We also report character-level BLEU scores (Papineni et al., 2002) using SacreBLEU (Post, 2018) as a supplement. For CFQ, we use exact match accuracy to evaluate model performance, where natural language utterances are mapped to meaning representations.

### 4.2 Model Settings

**Machine Translation.** We compare our method with previous competitive systems: (1) Transformer (Vaswani et al., 2017): first proposes a new

---

[2] https://github.com/fxsjy/jieba
[3] https://github.com/moses-smt/mosesdecoder/blob/master/scripts/tokenizer/tokenizer.perl

[4] https://github.com/facebookresearch/fairseq/blob/main/examples/roberta/multiprocessing_bpe_encoder.py

| Model | MCD1 | MCD2 | MCD3 | Mean |
|---|---|---|---|---|
| LSTM+attention | 28.9 | 5.0 | 10.8 | 14.9 |
| Transformer | 34.9 | 8.2 | 10.6 | 17.9 |
| Universal Transformer | 37.4 | 8.1 | 11.3 | 18.9 |
| Evolved Transformer | 42.4 | 9.3 | 10.8 | 20.8 |
| CGPS | 13.2 | 1.6 | 6.6 | 7.1 |
| NSEN | 5.1 | 0.9 | 2.3 | 2.8 |
| T5-11B | 61.4 | 30.1 | 31.2 | 40.9 |
| T5-11B-mod | 61.6 | 31.3 | 33.3 | 42.1 |
| RoBERTa | 60.6 | 33.6 | 36.0 | 43.4 |
| HPD | 72.0 | **66.1** | **63.9** | **67.3** |
| Dangle | **78.3** | 59.5 | 60.4 | 66.1 |
| RoBERTa+CReg | 74.8 | 53.3 | 58.3 | 62.1 |
| COMPOSITION | 72.8 | 53.2 | 52.2 | 59.4 |

Table 2: Exact-match accuracy on different MCD splits of CFQ. Results are averaged over 3 random runs.

| Model | Alleviate E | K, V | $CTER_{Inst}$ ↓ | $CTER_{Aggr}$ ↓ |
|---|---|---|---|---|
| Transformer | ✗ | ✗ | 28.4% | 62.9% |
| COMPOSITION | ✓ | ✗ | 22.6% (-5.8%) | 55.1% (-7.8%) |
| COMPOSITION | ✓ | ✓ | **20.4% (-8.0%)** | **52.0% (-10.9%)** |

Table 3: CTERs (%) against alleviating E or K,V on the CG-test set, where $CTER_{Inst}$ and $CTER_{Aggr}$ denote instance-level and aggregate-level CTER respectively. E and K, V denote encoder and keys, values entanglement problem respectively.

encoder-decoder architecture based solely on attention mechanisms; (2) Transformer-Rela: only replaces sinusoidal (absolute) positional embedding with a relative one; (3) Transformer-Small: only decreases the number of encoder layers and decoder layers to 4, 4 respectively; (4) Transformer-Deep: only increases the number of encoder layers to 8; (5) Bow (Raunak et al., 2019): uses bag-of-words pre-training to improve the representation of the encoder upmost layer; (6) SeqMix (Guo et al., 2020a): synthesizes examples to encourage compositional behavior; (7) Dangle (Zheng and Lapata, 2022a): adaptively re-encodes (at each time step) the source input to disentangle the representation of the encoder upmost layer;[5] (8) Proto-Transformer (Yin et al., 2022): integrates prototypes of token representations over the training set into the source encoding to achieve the goal of categorization; (9) Transformer+CReg (Yin et al., 2023): promotes representation consistency across samples and prediction consistency for a single sample; (10) R-Dangle$_{sep}$ (Zheng and Lapata, 2022b): disentangles their representations and only re-encode keys periodically, at some interval; (11) DLCL (Wang et al., 2019): proposes an approach based on dynamic linear combination of layers (DLCL), and is one of the very popular EnocderFusion work. Our method is built on top of (1)-(4), i.e., COMPOSITION, COMPOSITION-Rela, COMPOSITION-Small and COMPOSITION-Deep. We also provide reasons for experiments on CoGnition without language models (see Appendix E).

**Semantic Parsing.** We compare our method with previous competitive systems: (1) LSTM + atten-

---

[5]We use the same variant reported by Zheng and Lapata (2022a) (i.e., Dangle-EncDec (abs)) with sinusoidal (absolute) positional embedding.

tion: introduces attention mechanism (Bahdanau et al., 2015) in LSTM (Hochreiter and Schmidhuber, 1997); (2) Transformer (Vaswani et al., 2017); (3) Universal Transformer (Dehghani et al., 2019): combines the parallelizability and global receptive field of feed-forward sequence models like the Transformer with the recurrent inductive bias of RNNs; (4) Evolved Transformer (So et al., 2019): uses wide depth-wise separable convolutions in the early layers of both the encoder and decoder; (5) CGPS (Li et al., 2019): leverages prior knowledge of compositionality with two representations, and adds entropy regularization to the encodings; (6) NSEN (Freivalds et al., 2019): is derived from the Shuffle-Exchange network; (7) T5-11B (Raffel et al., 2020): treats every natural language processing task as a text-to-text problem, and is therefore suitable for the semantic parsing tasks. T5-11B is a T5 model with 11B parameters finetuned on CFQ; (8) T5-11B-mod (Furrer et al., 2020): shows that using masked language model (MLM) pre-training together with an intermediate representation leads to significant improvements in performance; (9) RoBERTa (Liu et al., 2019): makes use of the RoBERTa-base model as the encoder and the randomly initialized Transformer decoder trained from scratch, where we use the same experimental settings of (Zheng and Lapata, 2022a); (10) HPD (Guo et al., 2020b): proposes a novel hierarchical partially ordered set (poset) decoding paradigm, which consists of three components: sketch prediction, primitive prediction, and traversal path prediction; (11) Dangle (Zheng and Lapata, 2022a); (12) RoBERTa+CReg (Yin et al., 2023); (13) COMPOSITION: builds on (9) with our method.

### 4.3 Results on CoGnition

The main results on CoGnition are shown in Table 1. We observe that: **(1)** COMPOSITION gives **20.4%** $CTER_{Inst}$ and **52.0%** $CTER_{Aggr}$, with a

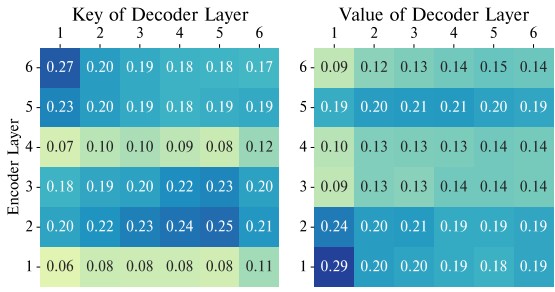

Figure 4: Learned composition weights (after normalized) that each encoder layer ($y$-axis) attending to keys or values of different decoder layers ($x$-axis).

significant improvement of **8.0%** and **10.9%** accordingly compared to the Transformer. Moreover, COMPOSITION significantly outperforms most baseline models under the almost same parameter settings,[6] indicating composing the syntactic and semantic information of sequences dynamically for a particular task is more beneficial to CG. Although Transformer+CReg achieves slightly better performance and contains fewer parameters, it is more complex and costly compared with COMPOSITION; **(2)** COMPOSITION, COMPOSITION-Rela, COMPOSITION-Small and COMPOSITION-Deep can deliver various performance improvements, demonstrating the general effectiveness of our method; **(3)** COMPOSITION-Deep performs better than Bow, Dangle and Proto-Transformer, indicating that focusing on alleviating the encoder entanglement problem only can achieve part of goals of CG as mentioned in Section 1. Compared to SeqMix, the improvement of COMPOSITION is more significant (2.3% vs 10.9% CTER$_{Aggr}$). SeqMix utilizes linear interpolation in the input embedding space to reduce representation sparsity, and we suppose that the samples synthesized randomly may be unreasonable and harmful to model training; **(4)** It can be seen that Transformer is even slightly better than DLCL, indicating DLCL and COMPOSITION do not share the same motivation or scenario.

### 4.4 Results on CFQ

The main results on CFQ are presented in Table 2. We observe that: **(1)** RoBERTa is comparable to T5-11B, T5-11B-mod and outperforms other baseline systems without pre-training except HPD, indicating that pre-training indeed benefits CFQ; **(2)** COM-

---

[6]We implement our approach based on Transformer-Deep for a fair comparison with Proto-Transformer.

| Model | CTER$_{Inst}$ ↓ | CTER$_{Aggr}$ ↓ |
|---|---|---|
| Transformer | 28.4% | 62.9% |
| COMPOSITION-SA | 22.2% (-6.2%) | 53.8% (-9.1%) |
| COMPOSITION-FF | 22.6% (-5.8%) | 55.6% (-7.3%) |
| COMPOSITION-SA & FF | **20.4% (-8.0%)** | **52.0% (-10.9%)** |

Table 4: CTERs (%) against composing different source information on the CG-test set.

POSITION substantially boosts the performance of RoBERTa (**43.4 → 59.4**), about **37%** relative improvements, and is in fact superior to T5-11B and T5-11B-mod. It also outperforms other baseline systems without pre-training except HPD. This result demonstrates that pre-training as a solution to CG also has limitations, and also indicates that COMPOSITION is complementary to pre-trained models; **(3)** HPD performs better than Dangle, RoBERTa+CReg and COMPOSITION, achieving 67.3 exact match accuracy, which is highly optimized for the CFQ dataset. On the contrary, COMPOSITION, RoBERTa+CReg and Dangle are generally applicable to any seq2seq models for solving any seq2seq tasks including MT, as mentioned in Section 4.3. However, compared with competitive performance on CoGnition, the improvements brought by COMPOSITION is relatively moderate, and even worse than Dangle. The underlying reason is related to a recent finding that compositionality in natural language is much more complex than the rigid, arithmetic-like operations (Li et al., 2021; Zheng and Lapata, 2022a; Dankers et al., 2022). MT is paradigmatically close to the tasks typically considered for testing compositionality in natural language, while our approach is more suitable for dealing with such scenarios.

## 5 Analysis

In this section, we conduct in-depth analyses of COMPOSITION to provide a comprehensive understanding of the individual contributions of each component. For all experiments, we train a COMPOSITION (6-6 encoder and decoder layers) instead of other experimental settings on the CoGnition dataset, unless otherwise specified.

### 5.1 Effects of Specific Keys and Values of Different Decoder Layers

As mentioned in Section 1 and 3.2, we hypothesize that keys, values entanglement problem ex-

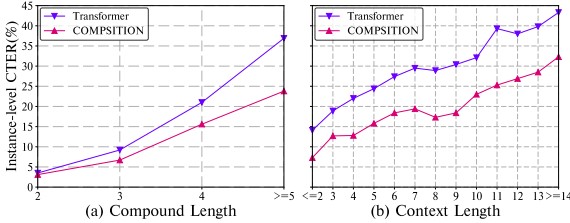

(a) Compound Length      (b) Context Length

Figure 5: CTER$_{Inst}$ of COMPOSITION and Transformer over the different compound and context lengths.

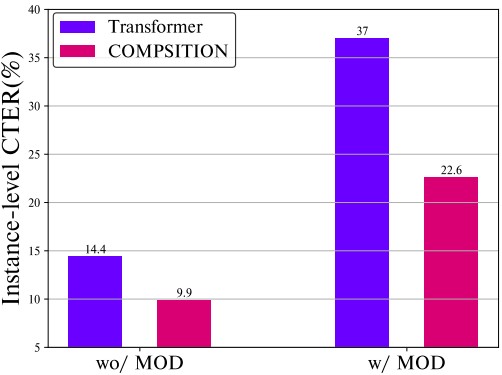

Figure 6: CTER$_{Inst}$ on compounds w/o and w/ MOD.

ists.[7] It is clear that our hypothesized keys, values entanglement problem is an extension to encoder entanglement problem. We show curiosity about whether this problem exists, and COMPOSITION can alleviate it. In this experiment, we investigate its influence on CoGnition. As shown in Table 3, we observe certain improvements (**-5.8%** and **-8.0%** CTER$_{Inst}$, **-7.8%** and **-10.9%** CTER$_{Aggr}$) when separately alleviating the encoder or keys, values entanglement problem.[8] It suggests that our method can alleviate both problems separately, and learning to compose information of different encoder layers dynamically can improve CG performance. Furthermore, the improvement brought from alleviating keys, values entanglement problem is more significant than that brought from alleviating encoder entanglement problem (52.0% vs 55.1% CTER$_{Aggr}$), demonstrating the reasonableness of keys, values entanglement problem.

To further illustrate the reasonableness of keys, values entanglement problem and understand how COMPOSITION alleviates it, we visualize the learned composition weights of COMPOSITION after normalized.[9] Specifically, we train COMPOSITION on CoGnition and then extract $W_k^i, W_v^i$ (see Section 3.2.2) to visualize them. Ideally, each key or value of different decoder layers should pay different attention weights to different encoder layers' information. As shown in Figure 4, the learned composition weights (after normalized) are mutually distinct for keys and values of different decoder layers, which implies COMPOSITION learns different dynamic composition modes for keys and values of every decoder layer respectively. In addition, it also indicates the reasonableness of keys, values entanglement problem we proposed, since keys and

---

[7]It is noteworthy that the representation of the encoder upmost layer serves as the same key and value passing into every decoder layer in the Transformer.

[8]We use one or $2N$ learnable vectors to generate one or $2N$ representations passing into $N$ decoder layers.

[9]We only use representations outputted by Eq. 6 for brevity.

values of different decoder layers utilize more than just the information of the encoder topmost layer. More importantly, it also emphasizes our method provides an effective composition of syntactic and semantic information, i.e., a specific combination for a particular task. To further demonstrate it, we also provide a toy experiment in Appendix C.

## 5.2 Effects of Composing Information of Encoder Layers or Sub-layers

As mentioned in Section 3, the Transformer encoder layer consists of two sub-layers. We assume that sub-layers may contain language information in different aspects, which may produce better generalization results. Therefore, we are curious about whether composing different encoder layers' or sub-layers' information is more beneficial to CG. In this experiment, we investigate its influence on CoGnition. Specifically, we train COMPOSITION to compose representations outputted by either Eq. 5 or 6 or a combination of both dynamically. Results are presented in Table 4. We observe certain improvements (**-6.2%** and **-5.8%** CTER$_{Inst}$) when separately composing SA- and FF-level representations, where SA and FF denote representations outputted by Eq. 5 and 6 respectively. Furthermore, the combination of both them brings further improvement (**-8.0%** CTER$_{Inst}$), which illustrates that the information in different encoder sub-layers is complementary and has cumulative gains. It also suggests that syntactic and semantic information brought by SA or FF is similar, but slightly different (Li et al., 2020), and can improve generalization performance respectively. It can be seen that the results of COMPOSITION-SA and COMPOSITION-FF presented in Table 4 are basically the same, and the improvements brought by the combination of both them is relatively moderate.

| Source | Transformer | COMPOSITION |
|---|---|---|
| **The waiter he liked** wore each other's clothes. | 他喜欢穿对方的衣服。
(**He liked** to wear each other's clothes.) | 他喜欢的服务员穿着彼此的衣服。
(**The waiter he liked** wore each other's clothes.) |
| **The waiter he liked** came by and chased the bully off. | 服务员来了，把那个恶霸赶走了。
(**The waiter** came by and chased the bully off.) | 他喜欢的服务员过来把那个恶霸赶走了。
(**The waiter he liked** came by and chased the bully off.) |
| **The waiter he liked** picked up his mail. | 服务员喜欢拿起他的邮件。
(**The waiter liked** to pick up his mail.) | 他喜欢的服务员拿起了他的邮件。
(**The waiter he liked** picked up his mail.) |

Table 5: Example translations of Transformer vs COMPOSITION. **The bold characters** denote the novel compounds and corresponding translations.

## 5.3 Effects on Compositional Generalization

**Compound Length and Context Length.** Longer compounds have more complex semantic information and longer contexts are harder to comprehend, making them more difficult to generalize (Li et al., 2021). We classify the test samples by compound length and context length, and calculate the $CTER_{Inst}$. In Figure 5, we can observe that COMPOSITION generalizes better as the compound and context grows longer compared to Transformer. In particular, COMPOSITION gives a lower CTER by **11.0%** over samples when the context length is more longer than 13 tokens. It suggests that our approach can better captures the compositional structure of human language.

**Complex Modifier.** The postpositive modifier atom (MOD) is used to enrich the information of its preceding word (e.g., *he liked* in the phrase *lost the dog he liked*), which is challenging to translate due to word reordering from English to Chinese. We divide the test samples into two groups according to compounds with (w/) or without (wo/) MOD. In Figure 6, we observe that the advantage of COMPOSITION grows larger in translating the compounds with MOD, demonstrating its superiority in processing complex semantic composition.

**Case Study.** We present 3 source examples containing a novel compound *the waiter he liked* with MOD and 4 atoms, and their translations in Table 5. For all samples, correct translations denote that the novel compounds are translated correctly. COMPOSITION correctly translates the novel compounds across different contexts for all samples, while Transformer suffers from omitting different atoms. For example, the translation of *the waiter* is omitted in the first example, *he liked* is omitted in the second example and *he* is omitted in the third example. Our results not only contain the correct compound translations but also achieve better translation quality, while Transformer makes errors on unseen compositions, confirming the necessity of composing the syntactic and semantic representations of sequences dynamically.

## 6 Conclusion

In this paper, we examine CG from a new perspective, i.e., utilizing different encoder layers' information. Specifically, we propose an extension to seq2seq models which composes different encoder layers' representations dynamically to generate specific keys and values passing into different decoder layers. Experiments on CoGnition and CFQ have shown the effectiveness of our proposal on CG without any dataset or task-specific modification. To our knowledge, we are the first to point out a new representation entanglement problem and investigate how to utilize information of different encoder layers benefits CG, achieving promising results on two realistic benchmarks. We hope the work and perspective presented in this paper can inspire future related work on CG.

## Limitations

There are two limitations of our approach. Firstly, compared with competitive performance on CoGnition, the improvements brought by COMPOSITION on CFQ is relatively moderate, and even worse than some competitive methods. Hence, COMPOSITION is more suitable for tasks typically considered for testing compositionality in natural language. We strongly recommend researchers pay more attention to tasks evaluating compositionality on natural language. Meanwhile, we regard that designing a more general method that can improve generalization performance in both synthetic and natural scenarios is a promising direction to explore in the future. Secondly, our method is mostly applicable to any seq2seq models which adopt an encoder-decoder architecture instead of encoder-only or decoder-only architecture. However, the methodology of the proposed COMPOSITION is still rather general to any seq2seq models which

adopt any architecture, since we can use the randomly initialized encoder or decoder to constitute the encoder-decoder architecture.

## Acknowledgement

We thank all the anonymous reviewers for their insightful and valuable comments. This work is supported by National key R&D Program of China (Grant no.2022ZD0116101), the Key Support Project of NSFC-Liaoning Joint Foundation (Grant no. U1908216), and the Project of Research and Development for Neural Machine Translation Models between Cantonese and Mandarin (No. WT135-76).

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

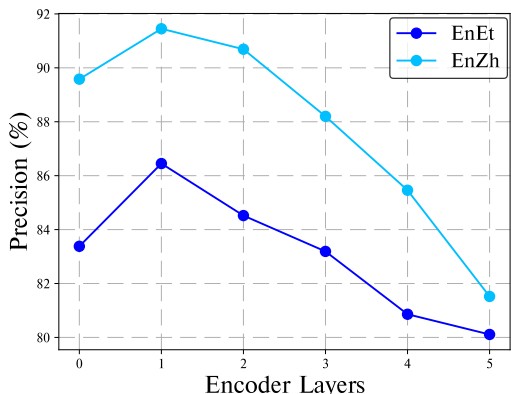

Figure 7: Precision (%) against different encoder layers' representations as input on the test set of POS tagging task.

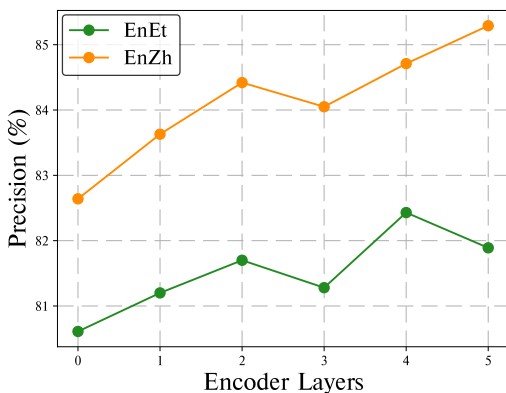

Figure 8: Precision (%) against different encoder layers' representations as input on the test set of Semantic tagging task.

Yafang Zheng, Lei Lin, Zhaohong Lai, Binling Wang, Shan Liu, Biao Fu, Wenhao Rao, Peigen Ye, Yidong Chen, and Xiaodong Shi. 2023. Layer-wise representation fusion for compositional generalization. *arXiv preprint arXiv:2307.10799*.

Denny Zhou, Nathanael Schärli, Le Hou, Jason Wei, Nathan Scales, Xuezhi Wang, Dale Schuurmans, Claire Cui, Olivier Bousquet, Quoc V Le, and Ed H. Chi. 2023. Least-to-most prompting enables complex reasoning in large language models. In *The Eleventh International Conference on Learning Representations*.

## A  Preliminary Analysis

In this section, we analyze the amount of syntactic and semantic information captured by different encoder layers in the Transformer under MT scenarios. We aim at analyzing the representations learned by different encoder layers of different models through probing the encoder as input representation for various prediction tasks. We measure the importance of input features for various tasks by evaluating the ability of the decoder. Specifically, we use a fixed encoder representation as input and two different tasks, i.e., Part-of-Speech (POS) tagging, and Semantic tagging, to evaluate the syntactic and semantic information contained in different encoder layers respectively. The reason is that we assume if the input representation effectively captures a property (syntactic or semantic information), then the decoder can easily predict that property.

To explore the precise effects of information captured by different encoder layers, we train the Transformer on the WMT18 English → Chinese (EnZh, rich-resource), English → Estonian (EnEt,

low-resource)[10] by following the same settings of Raganato et al. (2018).[11] After training the MT models, we freeze the encoder parameters, and only train one decoder layer[12] for each task, since we expect the decoder should not have overly significant impact on the model's performance of different tasks. We then analyze the amount of syntactic and semantic information in different encoder layers via evaluating the different encoder layers' performance of corresponding task. We use the Universal Dependencies English Web Treebank v2.0 (Zeman et al., 2017) for POS tagging (syntactic task) and the annotated data from the Parallel Meaning Bank (PMB) (Abzianidze et al., 2017) for Semantic tagging (semantic task).[13] We use precision to evaluate model performance.

Results on POS tagging and Semantic tagging are presented in Figure 7 and 8 respectively. We observe that:

- For EnEt and EnZh, the performance tends to decrease as the number of layers increase.

- For EnEt and EnZh, the performance tends to increase as the number of layers increase.

Therefore, we can conclude that **the bottom layers of the Transformer encoder contain more**

---

[10]We use two datasets with different sizes to analyze information captured by different encoder layers across models with different translation quality and target language.

[11]The provided datasets are freely available at `https://www.statmt.org/wmt18/translation-task.html`.

[12]We follow the same settings of Raganato et al. (2018) and adopt one attention head and one feed-forward sub-layer to consitute the decoder layer.

[13]We follow the same data preprocess process of Zeman et al. (2017) and Abzianidze et al. (2017).

| Model | CTER$_{Inst}$ ↓ | CTER$_{Aggr}$ ↓ |
|---|---|---|
| Transformer | 28.4% | 62.9% |
| Transformer-accu | failed | failed |
| COMPOSITION | **20.4% (-8.0%)** | **52.0% (-10.9%)** |

Table 6: CTERs (%) against Transformer-accu vs COMPOSITION on the CG-test set.

| Model | CTER$_{Inst}$ ↓ | CTER$_{Aggr}$ ↓ |
|---|---|---|
| Transformer | 28.4% | 62.9% |
| COMPOSITION-Half | 27.0% (-1.4%) | 61.3% (-1.6%) |
| COMPOSITION | **20.4% (-8.0%)** | **52.0% (-10.9%)** |

Table 7: CTERs (%) against Transformer, COMPOSITION and COMPOSITION-Half on the CG-test set.

**syntactic information and the top ones contain more semantic information**, and the information encoded by each encoder layer transforms from syntactic to semantic as the number of layers increase.

## B  Experimental Settings

For CoGnition, we set hidden size to 512 and feed-forward dimension to 1,024. The number of encoder and decoder layers are 6, 6 and the number of attention heads are 4. The model parameters are optimized by Adam (Kingma and Ba, 2015), with $\beta_1 = 0.9$, $\beta_2 = 0.98$. The learning rate is set to 5e-4 and the number of warm-steps is 4000. We set max tokens as 8,192 tokens for iteration. We use one GeForce GTX 2080Ti for training with 100,000 steps and decoding. We report the average performance over 6 random seeds provided in Li et al. (2021). We train all COMPOSITION models from scratch. For CFQ, we use the base RoBERTa with 12 encoder layers, which is combined with a Transformer decoder that has 2 decoder layers with hidden size 256 and feed-forward dimension 512. We use a separate target vocabulary. The number of attention heads are 8. The model parameters are optimized by Adam (Kingma and Ba, 2015), with $\beta_1 = 0.9$, $\beta_2 = 0.98$. The learning rate is set to 1e-4 and the number of warm-steps is 4000. We set max tokens as 4,096 tokens for iteration. We use one GeForce GTX 2080Ti for training with 45,000 steps and decoding. We report the average performance over 3 random seeds provided in Zheng and Lapata (2022a). We train COMPOSITION built on top of RoBERTa with full parameter fine-tuning.

## C  Effects of the Effective Composition

As mentioned in Section 3, we introduce the composed layer between the encoder and decoder to compose different encoder sub-layers' information dynamically to generate specific keys and values passing into different decoder layers. We show curiosity about whether the composed layer can fuse all encoder sub-layers' information effectively.

Therefore, we conduct a toy experiment on CoGnition. Specifically, all encoder sub-layers' information is accumulated to serve as the same key and value passing into every decoder layer (called Transformer-accu),[14] rather than composing them dynamically like we do. Results are listed in Table 6. Transformer-accu even fails to train. It suggests that even if the syntactic and semantic information of sequences is considered, the inappropriate combinations will instead bring noise to significantly affect the model's CG performance.

## D  Effects of Representations from Low-layer Encoder

To verify the low-layer encoder representations are also essential to our approach, we only evaluate our approach on CoGnition with the collected encoder representations of the top three layers. Results are presented in Table 7. We can observe that only composing the representations of the top three encoder layers leads to a sharp drop in performance (27.0% vs 20.4% CTER$_{Inst}$), but still outperforms the Transformer baseline (27.0% vs 28.4% CTER$_{Inst}$). It further demonstrates the distinct difference between our method and the findings introduced by previous studies on EncoderFusion. It also reflects our starting point is correct, i.e., exploring how to compose syntactic and semantic information. It can be seen that COMPOSITION's performance is dramatically reduced given only semantic information (the last three encoder layers' information).

## E  Reasons for Experiments on CoGnition without Language Models

We do not conduct experiments on CoGnition with language models for two reasons. **First**, CoGnition is constructed to test CG performance in MT scenarios with simple sentence pairs (see Figure 3), however, language models are trained on

---

[14] $\{y_0, ..., y_l\}$ are the output of the encoder layers $0 \sim l$. The input of keys and values of decoder layer $i$ is $x_i = y_0 + \cdots + y_{i-1}$, where $0 < i < L$.

vast amounts of multilingual sentences or bilingual sentence pairs. It is contrary to the compositional generalization task itself, since we can not guarantee that every sentence in the test set is a novel combination from known components for language models. **Second**, it is unfair to compare large language models with systems without pre-training. We strongly recommend researchers pay more attention to conduct experiments on CoGniton without language models.