# OpenReview forum: "Learning to Compose Representations of Different Encoder Layers towards Improving Compositional Generalization"
_EMNLP/2023/Conference — EMNLP 2023 Findings_

### Official Review · Reviewer_XNv7 · 2023-08-03

**Soundness:** 4

**Excitement:**

4: Strong: This paper deepens the understanding of some phenomenon or lowers the barriers to an existing research direction.

**Missing References:**

Basically, I think this work is novel and thus **no** new baseline work **must** be appended.

Just kindly note that there are a bunch of recent published works(e.g., in ACL'2023 proceedings) on compositional generalization which might be helpful for the authors to make the "Related Works" section up-to-date and thus make the audience better follow the recent progress of related works.

**Paper Topic And Main Contributions:**

This paper presents a method, dubbed "COMPOSITION", towards the improvement of compositional generalization and specifically, the disentanglement of the information (i.e., the keys and the values) passed between the encoder and the decoder in the transformer sequence-to-sequence model.

The proposed method is basically simple and reasonable, which is mainly introduced an additional module, composed layer, right between the encoder and the decoder, encouraging the decoder layers to consider the composition of the information outputted by each encoder layers rather than merely the upmost encoder layer (i.e., the original design for the transformer model). The authors also present the intuition of their design, which is to disentangle the semantic information and the syntactic information in the modeling of the natural language.

In the experiment part, they conduct experiments on two representative and challenging compositional generalization task, CoGnition(machine translation) and CFQ(semantic parsing), and give corresponding case studies to empirically demonstrate the effectiveness of "COMPOSITION" (results on CoGnition reach new **SOTA**.). Some necessary analytical results are also available in the paper to help further verify the reasonableness of the initial hypothesis.

**Questions For The Authors:**

A:In the paper, the authors metion the "source keys and values are entangled" in many places(e.g., line14-line16, line98-line99, and discussion on Fig.5). However, the detailed discussion on the entanglement of the source keys and values representations(i.e., what is on earth the entanglement of these representations? The authors provide a PCA visualization in Fig.5. Also, what does it represent when those representations are located in the same clusters?) is supposed to be further clarified in avoidance of puzzling readers.

B:The paper lacks the analysis on the extra(**compared with vanilla transformer-based models**) computation cost/ training time/ inference time brought by additionally introducing the composed layer and the different encoding representations passed to the different decoder layers. Could you please provide the analysis for this part?

C:It seems that "COMPOSITION" is also generic to the commonly-used pre-trained transformer-based model. Could you please provide the corresponding results of verifying the effectiveness of utilizing the method to boost the performance of more pre-trained models(e.g., T5 or BART) on the CoGnition task?

D:In the design of "COMPOSITION", what is the main difference between the $H_i^{SA}$ and the $H_i^{FF}$? Though the authors give the performance(Table.4) of separately using the $H_i^{SA}$ and the $H_i^{FF}$, I am wondering the reason why these two values are orthogonal to each other to some extent.(i.e., the reason why they both play roles in boosting the performance. e.g., are they represent different high-level info or something?).

**Reasons To Accept:**

(1):From my point of view, the proposed "COMPOSITION" is a quite generic method to improve the compositional generalization performance for most transformer-based seq2seq models, which is literally simple and thus requires only little modification on the original design of the transformer model.

(2):The paper is well-motivated and easy-to-follow in most places. Besides, the experimental results and the analysis can fully demonstrate the effectiveness of the proposed method (both in the task performance level, concrete case level (i.e., case studies) and in the learned representation level (i.e., analytical experiments in the **Section 4** and **Appendix**)). And hence I appreciate the rigorousness and the completeness of this paper, which are necessary ingredients of a good scientic paper.

**Reasons To Reject:**

This paper also has some weak points.

(1):The paper lacks the analysis on the extra(**compared with vanilla transformer-based models**) computation cost/ training time/ inference time brought by additionally introducing the composed layer and the different encoding representations passed to the different decoder layers.

(2):There are also some minor questions on the understanding of this paper. Please refer to the "Questions For The Authors" part (Note: the first weak point is also covered in that part, and hence it is OK for the authors to only respond to the questions in the "Question" part for their convenience).

**Reproducibility:**

4: Could mostly reproduce the results, but there may be some variation because of sample variance or minor variations in their interpretation of the protocol or method.

**Reviewer Confidence:**

4: Quite sure. I tried to check the important points carefully. It's unlikely, though conceivable, that I missed something that should affect my ratings.

**Typos Grammar Style And Presentation Improvements:**

line189: 2M learnable scalars? (I think here should be **vectors** ($H_i^{SA}$ or $H_i^{FF}$) instead of **scalars**?)

---

> ### Author Rebuttal · Authors · 2023-08-27
>
> ## Reply to Questions , Missing References and Typos Grammar Style And Presentation Improvements
> Thanks for your constructive comments and suggestions, and they are exceedingly helpful to improve our paper. We will incorporate them in the revised paper. In the following, your comments are first stated and then followed by our point-by-point responses.
> > Q1: In the paper, the authors metion the "source keys and values are entangled" in many places(e.g., line14-line16, line98-line99, and discussion on Fig.5). However, the detailed discussion on the entanglement of the source keys and values representations(i.e., what is on earth the entanglement of these representations? The authors provide a PCA visualization in Fig.5. Also, what does it represent when those representations are located in the same clusters?) is supposed to be further clarified in avoidance of puzzling readers.
>
> **A1**: Thank you very much for this insightful comment. As mentioned, the representation entanglement problem is the syntactic and semantic representations of sequences are twisted inappropriately. Previous work concentrates on **separating the learning of syntax and semantics, and then compose them appropriately**. On the contrary, we have conducted preliminary analysis provided in Appendix A, and concluded that **the bottom layers of the Transformer encoder contain more syntactic information and the top ones contain more semantic information**. Inspired by this, **we collect representations outputted by each encoder layer instead of separating the learning of syntax and semantics**. So one intuitive solution to solve the source keys, values entanglement problems is to learn **different and specific combinations** of syntactic and semantic information (i.e., representations outputted by each encoder layer) for keys and values of different decoder layers. Therefore, we propose a composed layer to compose them appropriately to generate specialized source keys and values passing into different decoder layers. As shown in Fig.5, in contrast to the Transformer, COMPOSITION obtains specialized keys and values representations passing into different decoder layers (i.e., **the dots representing keys and values of different decoder layers are in different places**). Notably, three circles represent randomly selected examples of CG-test set.
>
> > Q2: The paper lacks the analysis on the extra(compared with vanilla transformer-based models) computation cost/ training time/ inference time brought by additionally introducing the composed layer and the different encoding representations passed to the different decoder layers. Could you please provide the analysis for this part?
>
> **A2**: Thank you very much for this insightful comment. Following your suggestions, we have investigated its influence on CoGnition, and its results are presented as follows.
>
> |Models|upd/s|token/s|
> |---------|---------|---------|
> |Transformer|4.71(1.00$\times$)|5986.00(1.00$\times$)|
> |Dangle|1.23(0.26$\times$)|2505.58(0.42$\times$)|
> |COMPOSITION|4.05(0.86$\times$)|5728.01(0.7$\times$)|
>
> "upd/s" and "token/s" are abbreviated notations for "training updates per second" and "generated tokens per second". We observe that COMPOSITION keeps almost the same training and decoding speed as Transformer. On the contrary, Dangle prohibitively increases the training and decoding cost. Thus, it is clear that our method achieves promising results while brings little increase in computational cost compared with Transformer.
> > Q3: It seems that "COMPOSITION" is also generic to the commonly-used pre-trained transformer-based model. Could you please provide the corresponding results of verifying the effectiveness of utilizing the method to boost the performance of more pre-trained models(e.g., T5 or BART) on the CoGnition task?
>
> **A3**: Thank you very much for this insightful comment. We didn’t conduct experiments with commonly-used pre-trained transformer-based model (e.g., T5 or mBART) on CoGnition dataset for two reasons. **First**, it’s contrary to the compositional generalization task itself. Let’s back to the definition of compositional generalization task. Compositional generalization is the algebraic ability to understand and produce a potentially infinite number of **novel combinations** from **known components**, while PLM is a large language model, which are able to access **vast amounts of text data**. If we conduct experiments on PLMs to test their CG performance, one natural question can be raised: **how can we guarantee that every sentence in the test set is a novel combination from known components for LLMs**? **Second**, we want a fair comparison with previous work on CoGnition with a almost simliar parameter settings. \
> However, to validate the general effectiveness of our approach, we follow your suggestions and conduct experiments with mBART  (Liu et al., 2020) and our method built on top of mBART (called COMPOSITION-mBART) on CoGnition dataset. The full experiment results are presented below.
>
> | Models | Params | Instance-level CTER $\downarrow$ | Aggregate-level CTER $\downarrow$ |
> | --- | --- | --- | --- |
> | Transformer | 35M | 28.4% | 62.9% |
> | COMPOSITION| 35M | 20.4% | 52.0% |
> | mBART| 611M | 14.9% (-13.5%) | 40.0% (-22.9%) |
> | COMPOSITION-mBART| 611M | **13.9% (-14.5%)** | **37.5% (-25.4%)** |
>
> It can be seen that COMPOSITION-mBART outperforms mBART and achieves new state-of-the-art performance, demonstrating the general effectiveness of our method. However, compared with the impressive performance for COMPOSITION built on top of Transformer, the improvements brought by COMPOSITION-mBART is relatively moderate. The underlying reason is CoGnition is a synsestic, simple dataset where most sentences in the CG-test set are already seen in the PLMs' pre-training stage, causing less space for improving its CG performance. \
> Therefore, to further validate the effectiveness of our method, we have conducted experiments with COMPOSITION on COGS (Kim and Linzhen, 2020) dataset (one of representative semantic parsing benchmarks for compositional generalization). Specifically, we build our method on top of Transformer, and use initialized embeddings (on the both source and target side) with Glove (Pennington et al., 2014).  The results are reported below.
>
> | Models | Accuracy $\uparrow$ |
> | -- | -- |
> | Transformer | 85.5 |
> | Tree-MAML | 66.7 |
> | LexLSTM | 82.1 |
> | T5-base | 85.9|
> | COMPOSITION | **86.1** |
>
> It is clear that COMPOSITION outperforms most baseline models, indicating its general effectiveness on other compositional generalization tasks.
> > Q4: In the design of "COMPOSITION", what is the main difference between the $H_{SA}^{i}$ and $H_{FF}^{i}$? Though the authors give the performance (Table.4) of separately using the $H_{SA}^{i}$ and $H_{FF}^{i}$, I am wondering the reason why these two values are orthogonal to each other to some extent.(i.e., the reason why they both play roles in boosting the performance. e.g., are they represent different high-level info or something?).
>
> **A4**: Thank you very much for this insightful comment. $H_{SA}^{i}$ and $H_{FF}^{i}$ are the outputs of Transformer self-attention and feed-forward sub-layer $i$ (line 198-203), respectively. As mentioned in Appendix A, the information encoded by each encoder layer transforms from syntactic to semantic as the number of layers increase (line 1112-1115).
> The reason why they both play roles in boosting the performance and these two values are orthogonal to each other to some extent  is information brought by SA or FF is **similar**, but **slightly different** (Li et.al., 2020), respectively. It can be seen that the results of COMPOSITION-SA and COMPOSITION-FF presented in Table 4 are basically the same, and the improvements brought by the combination of both them is relatively moderate.
> > Q5: Just kindly note that there are a bunch of recent published works(e.g., in ACL'2023 proceedings) on compositional generalization which might be helpful for the authors to make the "Related Works" section up-to-date and thus make the audience better follow the recent progress of related works. line189: 2M learnable scalars? (I think here should be vectors ($H_{SA}^{i}$ or $H_{FF}^{i}$) instead of scalars?
>
> **A5**: Thank you so much for your careful reading! We will follow your suggestions and revise our paper for better readability.

---

### Official Review · Reviewer_5kLD · 2023-08-03

**Soundness:** 3

**Excitement:**

3: Ambivalent: It has merits (e.g., it reports state-of-the-art results, the idea is nice), but there are key weaknesses (e.g., it describes incremental work), and it can significantly benefit from another round of revision. However, I won't object to accepting it if my co-reviewers champion it.

**Missing References:**

Please check [LLM leaderboard](https://huggingface.co/spaces/HuggingFaceH4/open_llm_leaderboard) for models and corresponding references.

**Paper Topic And Main Contributions:**

In this paper, the authors proposed method COMPOSITION to improve the compositional generalization capability of transformers. The main idea is inspired by the findings that bottom layers of transformer contains more syntactic information while the top layers contain more semantic information. They proposed a composed layer that combines different encoder layers to improve CG. They conducted experiments on two datasets and demonstrated that the proposed approach outperforms compared baselines.

**Questions For The Authors:**

1. what is performance of other LLM look like on the two datasets?
2. what if the H^{SA} and H^{FF} have different dimensions (which is often the case in some LLMs)? Equation (7) and (8) are not applicable in this case.
3. what if we apply the listed encoder layer fusion methods on this task, what would be the performance like?

**Reasons To Accept:**

1. The studied problem, compositional generalization in machine translation, is important and critical to machine translation.
2. The proposed approach is simple and straightforward.
3. The paper is overall well-written and easy to follow.
4. The performance boost compared with listed baselines is promising.

**Reasons To Reject:**

1. The novelty of the proposed approach is limited. The difference between the proposed method and related encoder layer fusion approaches is marginal.
2. The compared models are pretty small. Many existing pretrained LLM are not included for comparison. Check [LLM leaderboard](https://huggingface.co/spaces/HuggingFaceH4/open_llm_leaderboard) for more details.
3. Some of the claims are not well justified, making the findings less solid. For example, line 141-143: "our proposed method is mostly applicable to any Seq2Seq models". The claim is not justified at all; line 335-337: "It indicates that large-scale language models, despite their huge number of parameters and training datasets, still fail to improve the CG performance". This claim is  sufficiently justified due to lack of evaluation.

**Reproducibility:**

4: Could mostly reproduce the results, but there may be some variation because of sample variance or minor variations in their interpretation of the protocol or method.

**Reviewer Confidence:**

4: Quite sure. I tried to check the important points carefully. It's unlikely, though conceivable, that I missed something that should affect my ratings.

---

> ### Author Rebuttal · Authors · 2023-08-27
>
> ## Reply to Questions and Reasons To Reject
> Thanks for your encouraging words and constructive comments. We sincerely appreciate your time in reading the paper, and our point-to-point responses to your comments are given below.
> > Q1: The novelty of the proposed approach is limited. The difference between the proposed method and related encoder layer fusion approaches is marginal.
>
> **A1**: Thank you very much for this insightful comment. We would like to address the novelty of this submission. There are three distinct differences between our method and EncoderFusion work. **Firstly**, our method exploits information from **all encoder sub-layers** and generates **specialized keys, values passing into different decoder layers** while they do not. **Secondly**, our method shows the effectiveness of **utilizing low-layer encoder representations** while they have the opposite view (see Appendix E). **Thirdly**, we do not share the same motivation or task. Their work focuses on how to transform information across layers in deep neural network scenarios. Our motivation is to learn to composing the syntactic and semantic representations of sequences appropriately for CG. We also provide the experimental results of DLCL (wang et.al., 2019) on the CoGnition dataset to further validate our above differences. Please refer to **A5** for the details.
> > Q2: The compared models are pretty small. Many existing pretrained LLM are not included for comparison. what is performance of other LLM look like on the two datasets?
>
> **A2**: Thank you very much for this insightful comment. We didn’t conduct experiments to evaluate different LLMs’ CG performance on CoGnition for two reasons. **First**, it’s hard to evaluate a broad range of LLMs’ CG performance on CoGnition due to time and resources limits. **Second**, it’s contrary to the compositional generalization task itself, and unfair to compare LLMs to systems without pre-training. Let’s back to the definition of compositional generalization task. Compositional generalization is the algebraic ability to understand and produce a potentially infinite number of **novel combinations** from **known components**, while LLM is a large language model, which are able to access **vast amounts of text data**. If we conduct experiments on LLMs to test their CG performance, one natural question can be raised: **how can we guarantee that every sentence in the test set is a novel combination from known components for LLMs**? \
> However, to validate LLMs' CG performance, we follow your suggestions and test GPT3.5 (also known as ChatGPT). We only choose GPT-3.5 (an effective and generic approach to address a wide range of NLP tasks) due to time and resource limits, and test its compositional generalization performance in the zero-shot manner on the CoGniton dataset. We didn't conduct experiments on CFQ dataset, because it’s hard to design the prompt as the input of GPT3.5 to generate results for CFQ. Specifically, we take *translate from English to Chinese: source sentence* as the input of GPT3.5 to generate the target sentence using greedy decoding. Finally, we obtain 10,554 generated examples, and 246 examples are discarded as a result of the error (i.e., The response was filtered due to the prompt triggering Azure OpenAI’s content management policy). The full experiment results are presented below.
>
> |Models|Instance-level CTER $\downarrow$|Aggregate-level CTER $\downarrow$|
> |---------|---------|---------|
> |Transformer|28.4%|62.9%|
> |GPT-3.5|28.1% (-0.3%)|71.9% (+9.0%)|
> |COMPOSITION|**20.4% (-8.0%)**|**52.0% (+10.9%)**|
>
> It can be seen that Transformer is even slightly better than GPT3.5, indicating GPT3.5 still fails to improve the CG performance. Meanwhile, there have some works investigating whether scaling up model size also improve compositional generalization (Qiu et.al., 2022; Hosseini et.al., 2022; Kim et.al., 2022). They found performance for the largest model size is generally worse than finetuning performance for much smaller models.
> > Q3: Some of the claims are not well justified, making the findings less solid. For example, line 141-143: "our proposed method is mostly applicable to any Seq2Seq models". The claim is not justified at all. what if the $H^{SA}$ and $H^{FF}$ have different dimensions (which is often the case in some LLMs)? Equation (7) and (8) are not applicable in this case.
>
> **A3**: Thank you very much for this insightful comment. Yes, you are right to some extent. if the $H^{SA}$ and $H^{FF}$ have different dimensions, our method are not applicable to compose all sub-layers’ information appropriately. However, the original design of our approach is to compose each encoder layer’s information instead of each encoder sub-layer’s information appropriately. The reason why we conduct this experiment is that we are curious about whether composing different encoder layers’ or sub-layers’ information is more beneficial to CG. Results in Table 4 also indicate that COMPOSITION-SA or COMPOSITION-FF can also bring certain improvements compared with Transformer. So the methodology of the proposed COMPOSITION is rather general to many Seq2Seq models.
> > Q4: It indicates that large-scale language models, despite their huge number of parameters and training datasets, still fail to improve the CG performance". This claim is sufficiently justified due to lack of evaluation.
>
> **A4**: Thank you very much for this insightful comment. We must admit this claim is insufficiently justified on CoGniton to some extent, because it's hard to evaluate a broad range of LLMs’ CG performance on CoGnition due to resource limits. However, there have some works investigating whether scaling up model size also improve compositional generalization (Qiu et.al., 2022; Hosseini et.al., 2022; Kim et.al., 2022). They found performance for the largest model size is generally worse than finetuning performance for much smaller models. We also provide experiments with the GPT3.5 on the CoGnition dataset in **A2** to further validate this argument. Finally ,we will revise these claims to make it clearer and more coherent.
> > Q5: what if we apply the listed encoder layer fusion methods on this task, what would be the performance like?
>
> **A5**: Thank you very much for this insightful comment. Following your suggestions, we reproduce DLCL (Wang et.al., 2019), one of the very popular EnocderFusion work, and conduct experiments on CoGnition dataset. The results are reported below.
>
> |Models|Instance-level CTER $\downarrow$|Aggregate-level CTER $\downarrow$|BLEU $\uparrow$|
> |---------|---------|---------|---------|
> |Transformer|28.4%|62.9%|59.5|
> |DLCL|28.4% (+0.0%)|67.9% (+5.0%)|59.2 (-0.3)|
> |COMPOSITION|**20.4% (-8.0%)**|**52.0% (+10.9%)**| **61.5 (+2.0)** |
>
> Results are averaged over 6 random runs on CoGnition. It can be seen that Transformer is even slightly better than DLCL, indicating DLCL and COMPOSITION do share the same motivation or scenario. DLCL focuses on how to transform information across layers in deep neural network scenarios, while COMPOSITION is to learn to composing the syntactic and semantic representations of sequences appropriately for CG.

---

### Official Review · Reviewer_VRuK · 2023-08-11

**Soundness:** 3

**Excitement:**

2: Mediocre: This paper makes marginal contributions (vs non-contemporaneous work), so I would rather not see it in the conference.

**Paper Topic And Main Contributions:**

This paper proposes a modification of the Transformer architecture, so as to allow keys and queries in decoder source attention sublayers to be redefined as weighted sums of encoder sublayers.
To motivate this modification, the authors frame their experiments with respect to the task of compositional generalization. They demonstrate the effectiveness of the proposed approach on two datasets (an MT dataset and a semantic parsing dataset).

**Reasons To Accept:**

- The authors report significant improvement on the synthetic test split of the MT dataset, and some improvement over their baseline encoder on the semantic parsing dataset.
- The solution put forth is conceptual rather simple, and allows for some interesting manual analyses (in particular, fig 4, lines 371 to 401)

**Reasons To Reject:**

**A/** The novelty in terms of architecture is limited, despite being pitched as a major point of interest of the paper. As the authors acknowledge (lines 550 sq.), there is significant overlap with prior work. It might have been more straightforward to not  present the composition layer as an entirely novel contribution, and instead devote some space to presenting prior work this approach builds upon  in §2.1 (esp. Wang et al 2019) clearly delineate in §2.2 how the proposed approach differ from this clearly very related prior work.

**B/** On a more fundamental level, the authors posit that transformer encoder representations "entangle" semantic and syntax, and claim that their proposal allows to mitigate this problem (lines 103 sq., e.g.). I find support for this claim to be minimal: there is no clear experiment showing that the learned representations disentangle semantic and syntactic factors in any way. Supposedly, appendix D discusses this aspect but only presents an inconclusive supplementary experiment.

~**C/** Some methodological details are unclear, deferred without notice to the appendix or absent. E.g., :~
 - ~I'm systematically uncertain across all experiments as to which of the following three approaches the authors  adopt: (i) fine-tuning a model with a limited number of added trainable parameters (ii) tuning only the newly-added learnable weight vectors, (iii) training a model from scratch~
 - ~Some of the baselines are fundamentally unknown to me: I don't know what "Transformer-rela" is and no reference is provided for me to look it up. No reference to appendix , where this term is defined, had been provided.~

**Edit after rebuttal:** some clarifications made by he authors greatly alleviated my concerns as per this last point.

**Reproducibility:**

3: Could reproduce the results with some difficulty. The settings of parameters are underspecified or subjectively determined; the training/evaluation data are not widely available.

**Reviewer Confidence:**

3: Pretty sure, but there's a chance I missed something. Although I have a good feel for this area in general, I did not carefully check the paper's details, e.g., the math, experimental design, or novelty.

**Typos Grammar Style And Presentation Improvements:**

I suggest adopting standard "matrix cookbook" notations for all equations. In particular, $W$ is generally reserved for weight matrices in ML literature, using this symbol to refer to a scalar is needlessly confusing.

---

> ### Author Rebuttal · Authors · 2023-08-27
>
> ## Reply to Reasons To Reject and Typos Grammar Style And Presentation Improvements
> We sincerely thank the reviewer for carefully reading our paper and pointing out our typos. We will carefully incorporate them in the revised paper. In the following, your comments are first stated and then followed by our point-by-point responses.
> > Q1: The novelty in terms of architecture is limited, despite being pitched as a major point of interest of the paper. As the authors acknowledge (lines 550 sq.), there is significant overlap with prior work. It might have been more straightforward to not present the composition layer as an entirely novel contribution, and instead devote some space to presenting prior work this approach builds upon in §2.1 (esp. Wang et al 2019) clearly delineate in §2.2 how the proposed approach differ from this clearly very related prior work.
>
> **A1**: Thank you very much for this insightful comment. We would like to address the novelty of this submission. There are three distinct differences between our method and EnocderFusion work. **Firstly**, our method exploits information from **all encoder sub-layers** and generates **specialized keys, values passing into different decoder layers** while they do not. **Secondly**, our method shows the effectiveness of utilizing **low-layer encoder representations** while they have the opposite view (see Appendix E). **Thirdly**, we do not share the same motivation or task. Their work focuses on how to transform information across layers in deep neural network scenarios. Our motivation is to learn to composing the syntactic and semantic representations of sequences appropriately for CG. Both our method and DLCL (Wang et.al., 2019) use learnabe vectors or scalars, however, the way we use them is completely different than they do. Importantly, we do not share the same motivation or task. Therefore, we would like to describe our method based on the findings conclued by our prelimary analysis (see Appendix A), not built upon in DLCL (Wang et.al., 2019). To further validate the above differences between our approach and EncoderFusion work, we reproduce DLCL (Wang et.al., 2019), one of the very popular EnocderFusion work, and conduct experiments on CoGnition dataset. The results are reported below.
>
> |Models|Instance-level CTER $\downarrow$|Aggregate-level CTER $\downarrow$|BLEU $\uparrow$|
> |---------|---------|---------|---------|
> |Transformer|28.4%|62.9%|59.5|
> |DLCL|28.4% (+0.0%)|67.9% (+5.0%)|59.2 (-0.3)|
> |COMPOSITION|**20.4% (-8.0%)**|**52.0% (-10.9%)**|**61.5 (+2.0)**|
>
> Results are averaged over 6 random runs on CoGnition. It can be seen that Transformer is even slightly better than DLCL, indicating DLCL and COMPOSITION do not share the same motivation or scenario. DLCL focuses on how to transform information across layers in deep neural network scenarios, while COMPOSITION is to learn to composing the syntactic and semantic representations of sequences appropriately for CG.
> > Q2: On a more fundamental level, the authors posit that transformer encoder representations "entangle" semantic and syntax, and claim that their proposal allows to mitigate this problem (lines 103 sq., e.g.). I find support for this claim to be minimal: there is no clear experiment showing that the learned representations disentangle semantic and syntactic factors in any way. Supposedly, appendix D discusses this aspect but only presents an inconclusive supplementary experiment.
>
> **A2**: Thank you very much for this insightful comment. As mentioned, the representation entanglement problem is the syntactic and semantic representations of sequences are twisted inappropriately. Previous work concentrates on **separating the learning of syntax and semantics, and then compose them appropriately**. On the contrary, we have conducted preliminary analysis provided in Appendix A, and concluded that **the bottom layers of the Transformer encoder contain more syntactic information and the top ones contain more semantic information**. Inspired by this, **we collect representations outputted by each encoder layer instead of separating the learning of syntax and semantics**. So one intuitive solution to solve the source keys, values entanglement problems is to learn **different and specific combinations** of syntactic and semantic information (i.e., representations outputted by each encoder layer). Therefore, we propose a composed layer to compose them appropriately to generate specialized source keys and values passing into different decoder layers. As shown in Fig.5, in contrast to the Transformer, COMPOSITION obtains specialized keys and values representations passing into different decoder layers (i.e., **the dots representing keys and values of different decoder layers are in different places**). Notably, three circles represent randomly selected examples of CG-test set.
>
> > Q3: I'm systematically uncertain across all experiments as to which of the following three approaches the authors adopt: (i) fine-tuning a model with a limited number of added trainable parameters (ii) tuning only the newly-added learnable weight vectors, (iii) training a model from scratch.
>
> **A3**: Thank you very much for this insightful comment. For CoGniton, we train COMPOSITION from scratch. For CFQ, we train COMPOSITION built on top of RoBERTa with full parameter finetuing.
>
> > Q4: Some of the baselines are fundamentally unknown to me: I don't know what "Transformer-rela" is and no reference is provided for me to look it up. No reference to appendix , where this term is defined, had been provided.
>
> **A4**: Thank you very much for this insightful comment. We have already mentioned in line 304-305 and Appendix C. Transformer-Rela is only replaces sinusoidal (absolute) positional embedding with a relative one.
>
> > Q5: I suggest adopting standard "matrix cookbook" notations for all equations. In particular, $W$is generally reserved for weight matrices in ML literature, using this symbol to refer to a scalar is needlessly confusing.
>
> **A5**: Thank you so much for your careful reading! We will follow your suggestions and revise our paper for better readability.

---

### Official Review · Reviewer_DCYU · 2023-08-11

**Soundness:** 3

**Excitement:**

3: Ambivalent: It has merits (e.g., it reports state-of-the-art results, the idea is nice), but there are key weaknesses (e.g., it describes incremental work), and it can significantly benefit from another round of revision. However, I won't object to accepting it if my co-reviewers champion it.

**Paper Topic And Main Contributions:**

This paper proposes COMPOSITION, a method that learns to compose representations from different layers of encoder, as a way to mitigate the issue of entangled syntactic and semantic problem in seq2seq models. The proposed method introduces a composed layer to generate keys and values for each decoder layer. The paper demonstrates that this approach improves compositional generalization on machine translation (CoGnition) and semantic parsing (CFQ) tasks. Finally, the paper provides detailed ablation studies and qualitative analysis.

**Questions For The Authors:**

A. Do the authors have results for the in distribution splits? Does adding the composed layer hurt the IID performance?

B. Figure 5, is there similar visualization for the standard Transformer for comparison?

C. Do the authors have mean and variance of several runs? for at least main experiments


**Reasons To Accept:**

- This paper introduces a simple approach that is generally applicable to any seq2seq model. The method does not require a significant change in architecture and demonstrates strong performance for compositional generalization on two tasks, especially in machine translation.
- The paper is well-motivated. The investigation of the syntactic and semantic entanglement problem is interesting. The related analysis is comprehensive.


**Reasons To Reject:**

- The in-distribution performance is not reported. It is unclear whether the proposed method favors the specific compositional generalization setups studied in this paper, but may not be generally applicable.
- The reported results are only one run. It would be better to include mean and variance from several runs, since most models are small size.


**Reproducibility:**

4: Could mostly reproduce the results, but there may be some variation because of sample variance or minor variations in their interpretation of the protocol or method.

**Reviewer Confidence:**

4: Quite sure. I tried to check the important points carefully. It's unlikely, though conceivable, that I missed something that should affect my ratings.

**Typos Grammar Style And Presentation Improvements:**

- Figure 4 seems to be hard to interpret, it might help readers to understand the visualization better by normalizing the weights
- L489-492 refers to qualitative examples in Table 6, which should be Table 5.
- I found Appendix A provides a nice motivation. The author could consider moving this section to the main text in the later version.

---

> ### Author Rebuttal · Authors · 2023-08-27
>
> ## Reply to Questions and Typos Grammar Style And Presentation Improvements
> Thanks for your constructive comments and suggestions, and they are exceedingly helpful to improve our paper. We will incorporate them in the revised paper. In the following, your comments are first stated and then followed by our point-by-point responses.
> > Q1: Do the authors have results for the in distribution splits? Does adding the composed layer hurt the IID performance?
>
> **A1**: Thank you very much for this insightful comment.  For CoGnition, we report BLEU on in-domain test set (ind-test). The detailed results are given below.
>
> |Models|ind-test $\uparrow$|
> | --- | --- |
> |Transformer|56.4|
> |COMPOSITION|**56.7 (+0.3)**|
>
> Results are averaged over 6 random runs on CoGnition. It is clear that improvements on compositional generalization are not at the expense of in-domain performance (COMPOSITION obtains similar and even slightly better performance than the Transformer on the CoGniton in-domain test set).
> > Q2: Figure 5, is there similar visualization for the standard Transformer for comparison?
>
> **A2**: Thank you very much for this insightful comment. For standard Transformer, the representation of the encoder upmost layer serves as the same key and value passing into every decoder layer. The detailed visualization is shown below in tabular form, where each encoder layer (y-axis) attending to keys of different decoder layers (x-axis).
>
> ||1|2|3|4|5|6|
> |---------|---------|---------|---------|---------|---------|---------|
> |**1**|0|0|0|0|0|0|
> |**2**|0|0|0|0|0|0|
> |**3**|0|0|0|0|0|0|
> |**4**|0|0|0|0|0|0|
> |**5**|0|0|0|0|0|0|
> |**6**|1|1|1|1|1|1|
>
> We omit visualization for each encoder layer attending to values of different decoder layers for brevity, since it is the same as the above table.
> > Q3: Do the authors have mean and variance of several runs? for at least main experiments.
>
> **A3**: Thank you very much for this insightful comment. We have already mentioned in line 263-264 and provided more details in Appendix B. For CoGnition, we report the average performance over 6 random seeds provided in Li et al. (2021). For CFQ, we report the average performance over 3 random seeds provided in Zheng and Lapata (2022). We also provide experimental results for each random seed, the detailed results are as follows.
>
> |CoGnition (seed)|Instance-level CTER|Aggregate-level CTER|BLEU|
> |---------|---------|---------|---------|
> |1|20.2%|52.1%|61.7|
> |7|20.5%|53.2%|61.4|
> |77|19.7%|50.7%|61.9|
> |777|18.4%|49.9%|61.8|
> |7777|21.1%|52.5%|61.3|
> |77777|22.5%|53.4%|60.9|
>
> |CFQ (seed)|MCD1|MCD2|MCD3|Mean|
> |---------|---------|---------|---------|---------|
> |1| 74.7 | 53.3 | 52.2 |60.1|
> |7| 72.1 | 50.9 | 47.9 |57.0|
> |77| 71.7 | 55.3 | 56.4 |61.1|
>
> Therefore, for CoGnition, the mean values of instance-level CTER, aggregate-level CTER and BLEU are **20.4**, **52.0** and **61.5**, respectively, and their variances are **1.6**, **1.6** and **0.1**, respectively. For CFQ, the mean value and variance of exact-match accuracy on Mean MCD splits are **59.4** and **3.0**, respectively.
>
> > Q4: Figure 4 seems to be hard to interpret, it might help readers to understand the visualization better by normalizing the weights; L489-492 refers to qualitative examples in Table 6, which should be Table 5; I found Appendix A provides a nice motivation. The author could consider moving this section to the main text in the later version.
>
> **A4**: Thank you so much for your careful reading! We will follow your suggestions and revise our paper for better readability.

---

### Meta-Review · Area_Chair_Wfvc · 2023-09-17

**Recommendation:** 3

**Metareview:**

This paper proposes a layer which composes the representations produced by the different layers of the encoder and outputs keys and values which are fed into the different layers of the decoder. The proposed layer is evaluated in the tasks of machine translation and semantic parsing.

The paper received mixed reviews. The reviewers found that that the paper has some merits: (i) the proposed approach is simple and quite general; (ii) the method appears to be empirically strong in the machine translation task; (iii) the considered problem is interesting and deserves further research. However, the reviewers also raised various concerns mainly about the lack of novelty, the lack of large baselines and about claims that appear in the paper and which are not well-justified. For instance, one reviewer stressed that there is no evidence that the proposed method indeed effectively combines semantic with syntactic information. There were also concerns about the presentation of the paper which gives the reader the impression that the proposed method is entirely novel and that there is no overlap with prior work.

Taking author response into account, I think that the paper is interesting and that this line of investigation is worthwhile, but I also agree with some of the concerns that were raised in the reviews.

---

### Decision · Program_Chairs · 2023-10-07

**Decision:**

Accept-Findings

**Comment:**

This paper proposes a layer which composes the representations produced by the different layers of the encoder and outputs keys and values which are fed into the different layers of the decoder. The proposed layer is evaluated in the tasks of machine translation and semantic parsing.

The paper received mixed reviews. The reviewers found that that the paper has some merits: (i) the proposed approach is simple and quite general; (ii) the method appears to be empirically strong in the machine translation task; (iii) the considered problem is interesting and deserves further research. However, the reviewers also raised various concerns mainly about the lack of novelty, the lack of large baselines and about claims that appear in the paper and which are not well-justified. For instance, one reviewer stressed that there is no evidence that the proposed method indeed effectively combines semantic with syntactic information. There were also concerns about the presentation of the paper which gives the reader the impression that the proposed method is entirely novel and that there is no overlap with prior work.

Taking author response into account, I think that the paper is interesting and that this line of investigation is worthwhile, but I also agree with some of the concerns that were raised in the reviews.